# Genomic Regions Associated with Resistance to Three Rusts in CIMMYT Wheat Line “Mokue#1”

**DOI:** 10.3390/ijms241512160

**Published:** 2023-07-29

**Authors:** Naeela Qureshi, Ravi Prakash Singh, Blanca Minerva Gonzalez, Hedilberto Velazquez-Miranda, Sridhar Bhavani

**Affiliations:** 1International Maize and Wheat Improvement Center (CIMMYT), Carretera Mexico-Veracruz Km. 45, El-Batan, Texcoco 56237, Mexico; n.qureshi@cgiar.org (N.Q.); r.singh@cgiar.org (R.P.S.); b.gonzalez@cgiar.org (B.M.G.); h.velazquez@cgiar.org (H.V.-M.); 2International Maize and Wheat Improvement Center (CIMMYT), ICRAF Campus, United Nations Avenue, Gigiri, Nairobi P.O. Box 1041-00621, Kenya

**Keywords:** genetic analysis, pleiotropic loci, stripe rust, leaf rust, stem rust, QTL mapping

## Abstract

Understanding the genetic basis of rust resistance in elite CIMMYT wheat germplasm enhances breeding and deployment of durable resistance globally. “Mokue#1”, released in 2023 in Pakistan as TARNAB Gandum-1, has exhibited high levels of resistance to stripe rust, leaf rust, and stem rust pathotypes present at multiple environments in Mexico and Kenya at different times. To determine the genetic basis of resistance, a F_5_ recombinant inbred line (RIL) mapping population consisting of 261 lines was developed and phenotyped for multiple years at field sites in Mexico and Kenya under the conditions of artificially created rust epidemics. DArTSeq genotyping was performed, and a linkage map was constructed using 7892 informative polymorphic markers. Composite interval mapping identified three significant and consistent loci contributed by Mokue: *QLrYr.cim-1BL* and *QLrYr.cim-2AS* on chromosome 1BL and 2AS, respectively associated with stripe rust and leaf rust resistance, and *QLrSr.cim-2DS* on chromosome 2DS for leaf rust and stem rust resistance. The QTL on 1BL was confirmed to be the *Lr46/Yr29* locus, whereas the QTL on 2AS represented the *Yr17/Lr37* region on the 2NS/2AS translocation. The QTL on 2DS was a unique locus conferring leaf rust resistance in Mexico and stem rust resistance in Kenya. In addition to these pleiotropic loci, four minor QTLs were also identified on chromosomes 2DL and 6BS associated with stripe rust, and 3AL and 6AS for stem rust, respectively, using the Kenya disease severity data. Significant decreases in disease severities were also demonstrated due to additive effects of QTLs when present in combinations.

## 1. Introduction

Wheat is the second-largest consumed cereal globally and is grown on over 219 million hectares, on all continents, with production of 778.6 million metric tons. It forms an important part of the human diet, providing more than 20% of daily calories and 15% of protein (FAO STAT 2020). Despite continuous breeding progress resulting in enhanced productivity and genetic gains, a range of biotic and abiotic stresses continue to challenge the production and productivity of wheat. Among major diseases, rust pathogens (stem rust, stripe rust, and leaf rust) are the most devastating and are present in all wheat-growing environments. These losses are further exacerbated by constant evolution, local, regional, and transboundary migration, as well as adaptation to changing climate. Widespread cultivation of genetically similar wheat varieties and improper use of race-specific resistance genes have also contributed to the pathogen resurgence in some regions. Estimated annual global losses due to wheat rust pathogens amount to around 15 million tons, valued at USD 2.9 billion [1]. 

Stem (or black) rust (SR) (caused by *P. graminis* f. sp. *tritici* (*Pgt*)), stripe (yellow) rust (YR) (*P. striiformis* f. sp. *tritici* (*Pst*)), and leaf (brown) rust (LR) (*P. triticina* (*Pt*)), occur in most wheat production environments, either singly or in combinations. However, the impact and scale/degree of damage varies based on host susceptibility, environmental suitability, and the pathogen’s virulence. Significant diversity exists within the three rust species and virulence variations are largely based on resistance genes deployed in the wheat varieties, which has often resulted in “Boom-and-Bust” cycles. Even though fungicides are widely used in some regions to control rusts, genetic resistance remains the most effective and environmentally friendly control strategy. To date, over 225 resistance genes that condition resistance to the three rust diseases have been officially designated [2,3]. Most of the characterized genes belong to the “race specific” gene class characterized by resistance effective for a single race, or a few races, of the pathogen, generally expressed at the seedling stage and mostly effective at all stages of the crop growth [4,5]. In contrast, some genes belonging to the race non-specific resistance class express at the adult plant stage conferring partial resistance, and often, in combination with other adult plant resistance (APR) genes, can lead to effective levels of durable resistance [6]. 

Of the catalogued YR resistance genes [2,3,7], the well-studied APR genes include *Yr11*, *Yr12*, *Yr13*, *Yr14*, *Yr16*, *Yr18*, *Yr29*, *Yr30*, *Yr36*, *Yr39*, *Yr46*, *Yr52*, *Yr59*, *Yr62*, *Yr68*, *Yr71*, *Yr75*, *Yr77*, *Yr78*, *Yr79*, *Yr80*, and *Yr82* for yellow rust (YR); *Lr12*, *Lr13*, *Lr22a*, *Lr34*, *Lr35*, *Lr46*, *Lr48*, *Lr49*, *Lr67*, *Lr68*, *Lr74*, *Lr75*, *Lr77*, and *Lr78* for leaf rust (LR); and *Sr2*, *Sr55*, *Sr56*, *Sr57*, and *Sr58* for stem rust (SR) resistance [2,3,8,9,10,11,12,13,14]. In addition to the permanently designated genes, several temporarily designated genes and quantitative trait loci (QTLs) have also been reported [8,15,16]. 

A key component in developing resistant wheat varieties involves identifying genes/QTLs and closely linked diagnostic markers that can facilitate deployment and accurate selection for the specific resistance gene(s). This process of discovery, characterization, and molecular mapping of resistance genes has been expedited with the recent development in the high throughput genotyping technologies such as Diversity Arrays Technology Sequencing (DArTSeq) [17] and 90K Infinium SNP chip array [18]. The availability of the complete wheat reference sequence has also become one of the important genomic tools for genetic studies [5,19]. 

“Mokue#1”, a CIMMYT breeding line, was distributed in 2017 through international trials and nurseries as 38th ESWYT (Elite Spring Wheat Yield Trial) entry#138, 50th IBWSN (International Bread Wheat Screening Nursery) entry#1158, and 11th SRRSN (Stem Rust Resistance Screening Nursery) entry# 6055. The average grain yield of Mokue#1 in trials conducted for three years in Mexico under optimally irrigated environments was 5% lower than that of the highest yielding check variety, Borlaug100, whereas under drought- and heat-stressed environments, its yield was 9% lower than that of Borlaug100. The average grain yield from 81 worldwide sites in the 38th ESWYT was 3% higher than the mean of the local checks and on par with the highest-yielding CIMMYT check. Mokue#1 showed exceptional resistance to multiple diseases; it was highly resistant to all three rusts in Mexico and all international sites according to the reported data, highly resistant to wheat blast, and moderately resistant to Septoria tritici blotch, spot blotch, and Fusarium head blight. It has very large, hard–semihard white grain, high test weight, high grain zinc and iron content, medium strong gluten, and good bread loaf volume. It possesses the following high and low molecular weight glutenin alleles: *Glu-A1*: 2, *Glu-B1*: 7 + 9, *Glu-D1*: 2 + 12, *Glu-A3*: c, *Glu-B3*: h, and *Glu-D3*: b, and it lacks the wheat–rye translocation 1BL.1RS. The presence of Glu-D1: 2 + 12 particularly makes it highly suitable for flat breads such as chapatti and tortilla. Mokue#1 was released in Pakistan in 2023 as “TARNAB Gandum-1”. This study was conducted to: (1) determine the genetic basis of YR, LR, and SR resistance in a “Mokue#1 × xApav#1” RIL population; (2) identify and map QTLs associated with rust resistance; and (3) develop closely linked molecular markers for marker-assisted breeding. 

## 2. Results

### 2.1. Genetic Analysis

#### 2.1.1. Seedling Rust Response Assessment

The resistant parent Mokue#1 showed infection type (IT) 23C (C = chlorosis) and the susceptible parent, Apav#1, was scored 4 when tested against YR pathotype (MEX14.191 and MEX16.04) on a 0–4 disease rating scale [12]. IT for the homozygous resistant (HR) RILs varied from 2C to 23CN (N = necrosis), whereas the homozygous susceptible (HS) RILs showed an IT of 4 (Figure 1A). The chi-squared test indicated single gene segregation with 118 homozygous resistant (HR) lines and 143 homozygous susceptible (HS) lines (χ^2^_1:1_ = 2.39, non-significant at *p* = 0.05 and 1 df). RILs that showed segregation were averaged and lines with an average disease score of more than 3 on the 0–4 scale were placed into susceptible categories, and lines with an average disease score of 3 or less were placed into resistant categories. The resistance gene was temporarily designated *YrMokue*.

For LR seedling assessment, Mokue#1 and Apav#1 showed ITs 23C and 4, respectively, against the *Pt* pathotype MBJ/SP. The chi-squared analysis of seedling response among the RIL population conformed to single gene segregation with 125 lines showing HR infection type and 136 lines showing HS infection type (χ^2^_1:1_ = 0.46, non-significant at *p* = 0.05 and 1 df). RILs that showed segregation were averaged and lines with an average disease score of more than 3 on the 0–4 scale were placed into susceptible categories, and lines with an average disease score of 3 or less were placed into resistant categories. The IT of HR RILs was 1C, while HS RILs showed an IT of 4 (Figure 1B), and the seedling gene was temporarily designated *LrMokue*.

#### 2.1.2. Field Rust Response Assessment

Mokue#1 showed an average YR severity of 8% with a moderately susceptible (MS) host reaction, whereas Apav#1 showed average severity of 92% with a susceptible (S) host response [20]. The frequency distribution appeared to be comparable across years (Figure 2a). High phenotypic correlations of 0.85–0.97 were observed between different datasets across years (Table 1).

For LR, Mokue showed an average severity of 2% with MSS (moderately susceptible to susceptible) host reaction, and Apav#1 showed disease scores of 90–100% with a susceptible host response (Figure 2b). Rust severity variation among the RIL population was similar among different years (Table 2). The phenotypic correlation for LR severity in the RIL population was between 0.69 and 0.94 (Table 2).

Phenotyping at Njoro, Kenya showed that Mokue#1 was highly resistant to both diseases with an average disease severity of 3% with R (resistant) host reaction for stripe rust and 2.5% severity with R-MR (resistant to moderately resistant) reactions for stem rust. In contrast, Apav#1 scored 70% and 85% for stripe rust and stem rust severity, respectively, with S (susceptible) host reactions. The frequency distribution (Figure 2c, d) showed a similar pattern across years with both stripe rust and stem rust in Kenya. High phenotypic correlations of 0.88–0.96 and 0.8–0.9 were observed for stripe rust and within-year stem rust severity data, respectively. However, across-year SR correlations were intermediate (0.48–0.59; Table 2). Quantitative variation for the traits was observed across years, showing its skewness towards the resistant group.

### 2.2. Linkage Map Construction

The Mokue#1/Apav#1 RIL population was genotyped using the DArTSeq platform established under SAGA at CIMMYT. Initially, 178,732 SNP markers were provided as raw data. Various filtering criteria were applied to remove redundant markers that were non-polymorphic, and with a lower than 70% call rate, greater than 20% heterozygosity, and over 20% missing data. Markers with more than 5% minor allele frequency were also discarded. After filtering out markers based on the above criteria, chi-squared analysis was carried out to avoid segregation distortion.

A high-density DArTSeq linkage map consisting of 7892 markers was constructed using the ASMap package in R software at a LOD threshold value of 6. The final Mokue#1/Apav#1 linkage map had 33 linkage groups, with the A genome represented by 2597 markers, the B genome with 3833 markers, and the D genome with 1462 markers (Figure 3), and an average marker density of 2–3 cM. 

### 2.3. QTL Mapping

#### 2.3.1. Seedling Resistance

*YrMokue* was mapped on chromosome 2AS between markers *100159996* and *WGGB156*, explaining 21% of phenotypic variation, whereas *LrMokue* was mapped on chromosome 2AS between markers *100043009* and *1229550*, explaining 6–8% of phenotypic variation.

#### 2.3.2. Pleiotropic QTL Identified for Adult Plant Resistance to Stripe Rust and Leaf Rust

Composite interval mapping identified two pleiotropic QTLs, *QLrYr.cim-1BL* and *QLrYr.cim-2AS,* for stripe rust and leaf rust in the Mokue#1/Apav#1 RIL population on the long arm of chromosome 1B and the short arm of chromosome 2A, respectively. Both QTLs were contributed by the resistant parent, Mokue#1. *QLrYr.cim-1B* and *QLrYr.cim-2AS* were consistently associated with stripe rust resistance (both in Mexico and Kenya) and leaf rust datasets across years in Mexico.

The 1BL QTL, *QLrYr.cim-1BL* explained 7–22% phenotypic variation and peaked between the markers *1202629a1* and *29307864a1* (Table 3 and Figure 4a). *QLrYr.cim-1BL* was mapped around 678-682 Mb of the physical assembly of Chinese Spring (CS) [19], which is 257–258 cM on the consensus map of wheat v4. The pleiotropic slow rusting gene *Lr46/Yr29/Sr58/Pm39* was also located on chromosome 1BL and, according to IWGSC (2018) physical assembly information of CS, this gene is located around 670–680 Mb. To further confirm *QLrYr.cim-1BL* to be *Lr46/Yr29/Sr58/Pm39,* RILs were also genotyped with closely linked markers *SNPG122* and *csLv46G122* and added to the linkage map. DArTSeq markers linked to *QLrYr.cim-1BL* were also converted into KASP markers (Table 4) and the entire population was genotyped. One in-house KASP marker, *CIM0003*, mapped close to the markers *SNPG122* and *csLv46G122*. All the KASP markers, *CIM0003*, *SNPG122*, and *csLv46G122* mapped between the DArTSeq markers *1202629a1* and *29307864a1*. The marker position on the consensus map, as well as the physical assembly of CS, along with the positioning of *SNPG122* and *csLv46G122* at the *QLrYr.cim-1BL* peak and the haplotype similarity between *Lr46/Yr29/Sr58/Pm39* markers and *CIM0003*, suggested this QTL is in fact *Lr46/Yr29/Sr58/Pm39*.

Another consistent locus was identified on chromosome 2AS, named *QYr.cim-2AS/QLr.cim-2AS*, which was contributed by Mokue#1. The race-specific resistance genes *Yr17*, *Lr37*, and *Sr38* are located on 2NS/2AS translocation, also known as VPM resistance. The entire 2NS/2AS translocation corresponds to about the 16 cM region on the distal short arm of the chromosome. *QLrYr.cim-2AS* mapped between DArTSeq markers *100159996* and *1696236*, which corresponds to around the 8 cM (3–6 Mb) region on the consensus map (Figure 4b), confirming it to be on the 2NS/2AS translocation. *QLrYr.cim-2AS* was consistently detected at highly significant levels, explaining 7–46% phenotypic variation (Table 3). However, this QTL was not effective against Kenyan *Pst* pathotypes. *Yr17/Lr37/Sr38* linked markers, *WGGB156*, *cslVrgal3*, and VPM SNP were also genotyped on the entire RIL population and added to the linkage map. Additionally, one in-house 2NS/2AS KASP marker (*CIM0001*) developed from another study was also tested on the population and was incorporated into the map (Table 4). All the KASP markers also mapped around the 8 cM of the 2NS/2AS translocation. Seedling data for stripe rust and leaf rust peaked at *QLrYr.cim-2AS*, confirming *YrMokue* to be *Yr17*. *Yr17* provides partial resistance at both the seedling and adult plant stage in Mexico. *Lr37* is ineffective at the seedling stage; however, the residual background effect or presence of a minor effect of the Lr-resistant gene at the same locus only accounts for 6–8% variation. Leaf rust APR data for the years 2019 and 2020 mapped the resistance QTL a bit further from all other datasets, between the DArTSeq markers *100043009* and *1229550* and explaining 15–19% of phenotypic variation, corresponding to 8–15 cM of the region (Figure 4b and Table 3). 

An additional consistent pleiotropic QTL for LR and SR resistance was also identified on the short arm of chromosome 2DS, *QLrSr.cim-2DS*. *QLrSr.cim-2DS* was consistently significant across the years for LR but was consistent in three replications for SR in 2021, explaining 7–20% of phenotypic variation. *QLrSr.cim-2DS* was mapped between DArTSeq markers *1102611* and *1111273*, corresponding to the 14–16 cM region on the consensus map (Table 3 and Figure 4c). About 37 of the linked DArTSeq markers were converted into KASP markers and only one KASP marker, *CIM0004*, showed a linkage and was genotyped on the entire RIL population and added to the linkage map (Table 4).

#### 2.3.3. Other Identified QTLs for Stripe Rust and Stem Rust Resistance

In Kenya, two additional minor but consistent YR QTLs were identified on chromosomes 2DL and 6BS: *QYrKen.cim-2DL* and *QYrKen.cim-6BS*, apart from the pleiotropic QTL *QLrYr.cim-1BL*. *QYrKen.cim-2DL* explained 8–15% and 4–8% of the phenotypic variation, respectively, and were mapped between the markers *100085243* and *100016397* for the years 2021 and 2020, and between *1391687* and *100043543* during 2020 (Table 3 and Figure 5a,b).

Two other minor but inconsistent QTLs for SR were also identified on chromosomes 3AL and 6AS, *QSr.cim-3AL* and *QSr.cim-6AS*, explaining 8–13% and 8–15% of the variation, respectively (Table 3 and Figure 5c, d). Both QTLs showed a significant association with the year 2020 datasets, whereas for *QSr.cim-6AS*, year 2021 data showed slight insignificance, but the peaks were observed. 

### 2.4. Additive Interactions of Different QTLs Enhancing Rust Resistance

The Mokue#1/Apav#1 RIL population was divided into different groups based on the presence of lines carrying an individual QTL, and those carrying QTLs in various combinations and lines without any QTL to assess their effect on disease severity. Generally, RILs carrying combinations of QTLs exhibited lower disease severity, indicating additive interactions of QTLs in enhancing disease resistance. RILs carrying *QLrYr.cim-1BL, QLrYr.cim-2AS*, and *QLrSr.cim-2DS* individually produced % mean LR severities of 21.9, 24.3, and 31.7, respectively. Lines carrying different dual combinations of QTLs showed lower mean disease severity as follows: *QLrYr.cim-1BL* + *QLrYr.cim-2AS* = 11.7%, *QLrYr.cim-1BL + QLrSr.cim-2DS* = 9%, and *QLrYr.cim-2AS + QLrSr.cim-2DS* = 13.5%. Mean disease severity of 4.4% for RILs carrying all three QTLs was significantly different from the other groups, whereas for RILs without any QTL, the mean disease severity was 64% (Figure 6a).

For YR, 34.8% and 9.2% mean disease severities were observed when *QLrYr.cim-1BL* and *QLrYr.cim-2AS*, respectively, were present individually. The combination of these QTLs resulted in lower disease severity of 4%, whereas in the absence of these QTLs, the mean disease severity was 64% in Mexico (Figure 6b).

For stripe rust in Kenya, the presence of a single QTL was associated with mean disease severities of 21.3% for *QLrYr.cim-1BL*, 44.6% for *QYrKen.cim-2DL*, and 36% for *QYrKen.cim-6BS*, whereas RILs that carried combinations of two QTLs exhibited 7% for *QLrYr.cim-1BL + QYrKen.cim-2DL*, 14% for *QLrYr.cim-1BL + QYrKen.cim-6BS*, and 23.7% for *QYrKen.cim-2DL + QYrKen.cim-6BS*. RILs that carried all three QTLs showed a lower mean YR severity of 10.8% (Figure 6c). 

RILs that carried *QLrSr.cim-2DS*, *QSr.cim-3AL*, and *QSr.cim-6AS* singly exhibited mean SR severities of 15.7%, 17.9%, and 20.5%, respectively, whereas two gene combinations showed 13.3% (*QLrSr.cim-2DS + QSr.cim-3AL*), 12% (*QLrSr.cim-2DS + QSr.cim-6AS*), and 14.7% (*QSr.cim-3AL + QSr.cim-6AS*). Mean disease severity of RILs with all three QTLs was significantly lower at 9.4% (Figure 6d).

## 3. Discussion

Continuous evolution of virulence in a pathogen population demands continuous efforts in identifying new genes conferring resistance and their deployment in breeding to achieve durable rust resistance to ensure stability of wheat varieties under production. The elite wheat line Mokue#1, developed by CIMMYT and released in Pakistan in 2023 as TARNAB Gandum-1, exhibited good levels of LR, YR, and SR resistance across multiple environments and years. We investigated the genetic basis of resistance to the three rust fungi in a Mokue#1/Apav#1 RIL population. Across the years, Mokue#1 showed high levels of resistance in both Mexico and Kenya, and quantitative variation in phenotypes in the RIL population suggested the presence of multiple genes/QTLs that had additive effects. We identified three major pleiotropic QTLs on chromosome 1BL (*QLrYr.cim-1BL*), 2AS (*QLrYr.cim-2AS*), and 2DS (*QLrSr.cim-2DS*); two consistent minor YR QTLs on chromosomes 2DL (*QYrKen.cim-2DL*) and 6BS (*QYrKen.cim-6BS*); and two minor inconsistent QTLs for SR resistance on chromosomes 3AL (*QSr.cim-3AL*) and 6AS (*QSr.cim-6AS*). 

### 3.1. QLrYr.cim-1BL

A pleiotropic QTL *QLrYr.cim-1BL* was mapped at 257–258 cM (~670–680 Mb) on the physical assembly of CS on chromosome 1BL. *QLrYr.cim-1BL* was consistent with most of the YR datasets from both Mexico and Kenya, and leaf rust datasets from Mexico across the years, explaining phenotypic variations of 8–22%, thereby showing its significance and stability in various environments. The chromosome 1BL is known to carry the slow rusting, pleiotropic adult plant gene *Lr46/Yr29/Sr58/Pm39* conferring resistance to multiple diseases [21]. The history of the 1BL region in the CIMMYT germplasm dates back to the early days of breeding, as several older semi-dwarf varieties, including “Pavon F76”, released from the CIMMYT wheat germplasm, possessed it [22,23]. *Lr46/Yr29/Sr58/Pm39* occurs frequently in the CIMMYT breeding program and is an important contributor of durable resistance against multiple diseases [24,25,26]. Over 90% of the CIMMYT breeding lines now possess this gene and its presence was reported in multiple genetic mapping studies involving the CIMMYT germplasm [27,28,29,30,31]. This gene is also associated with a morphological marker, leaf tip necrosis (LTN), in adult plants, which is also expressed in the resistant parent in all environments, suggesting that *QLrYr.cim-1BL* was in fact *Lr46/Yr29*. The physical position of *QLrYr.cim-1BL* in our study was the same as that reported earlier in different studies, which mapped *Lr46/Yr29* between 672.6 and 673.8 Mb on the physical assembly of CS [32]. To further confirm and validate *QLrYr.cim-1BL* to be *Lr46/Yr29*, different linked STS and KASP markers were also used in this study, which proved the presence of *Lr46/Yr29*. We also developed a new KASP marker linked to *Lr46/Yr29* (*QLrYr.cim-1BL*) named *CIM0003*, which mapped at the same interval as other reported linked markers and can be used in marker-assisted selection of *Lr46*/*Yr29*.

### 3.2. QLrYr.cim-2AS

The second QTL was identified on chromosome 2AS (*QLrYr.cim-2AS*) and based on its physical position of ~8 cM (6–7 Mb on physical assembly), thus confirming the presence of 2NS/2AS translocation, which was earlier reported at 25–38 cM [33]. However, with the first report of physical size estimation, the total translocation was estimated at 16 cM (32.6-33.5 Mb) [34]. The presence of 2N^V^S translocation from *Aegilops ventricosa* in the hexaploid wheat chromosome 2AS is very common, conferring resistance to multiple pests and diseases along with other beneficial traits [34]. This current 2NS/2AS translocation exhibited the largest phenotypic effect in reducing YR and LR severity. The 2NS/2AS translocation carrying wheat line VPM1 was found to carry race-specific resistance genes *Yr17*, *Lr37*, and *Sr38* [35]. A high frequency of 2N^V^S translocation both in the CIMMYT germplasm and multiple winter and spring winter breeding programs around the world has been documented [34]. Virulence for *Yr17* has been reported in most parts of the world [36,37,38]. Partial virulence was also observed in Mexico in 2007–2008 to *Pst*. The *Pst* pathotype MEX14.191 and *Pt* pathotype MBJ/SP used for seedling and field phenotyping carry virulence to *Yr17* and virulence to *Lr37*, respectively, when the standard YR and LR differential sets carrying *Yr17* in the near-isogenic background of “Avcoet” and *Lr37* in “Thatcher” backgrounds were used in seedling phenotyping. Mokue#1 and the non-segregating resistant lines in the population showed ITs 12C to 23C, which is typical for *Yr17* and often seen when it is present in combination with known pleiotropic APR loci and QTLs in the background. Several studies reported that variation in the expression of *Yr17* is largely influenced by genetic backgrounds and environmental conditions [39]. This QTL was later confirmed on the 2NS/2AS region through QTL mapping and validation using *Yr17/Lr37* linked markers. The *Ventruip/LN2* marker was tested on the parents; however, it was not mapped on the entire population owing to the dominant allele fingerprint, which could result in false positives. However, two STS markers, *WGGB156* [40] and *cslVrgal3* [41], and one KASP marker, *Yr17*/*Lr37*/*Sr58*, were used in the mapping study [42]. Alongside the above-reported markers, one in-house marker (*CIM0001*) designed in another study for 2NS/2AS translocation was also used in this study. All the markers peaked at *QLrYr.cim-2AS* QTL, which provided further evidence to conclude that this QTL was *Yr17*. *Yr17* is completely ineffective against the Kenyan *Pst* isolates and this complete virulence for *Yr17* has been reported since 2011 [37]. This is why no 2AS QTL was observed with the Kenyan dataset. The in-house marker *CIM0001*, along with the other 2NS/2AS markers, can be used in MAS for this *QTL*, or in detecting its presence in wheat lines. The *Pt* pathotype MBJ/SP used in the current study is known to carry virulence for *Lr37,* both at seedling and adult plant stages, indicating the presence of a new slow rusting gene on the same translocation based on its position.

### 3.3. QLrSr.cim-2DS

Another consistent pleiotropic QTL was detected on the short arm of chromosome 2D across all years for leaf rust and stem rust, explaining 14–24.7% phenotypic variation for LR and 6.4–8.7% for SR. The D genome is known to be always less polymorphic compared to the A and B genomes. Chromosome 2DS has eight cataloged race-specific resistance genes, including *Lr22a* derived from *Aegilops tauschii* and *Lr22b* derived from *Triticum aestivum*, conferring race-specific APR [12,43]. 

The APR QTL, *QLr.cim-2DS*, contributed by UC1110, was also reported on the top of the short arm of chromosome 2D [44], explaining 11.8–26.6% of the phenotypic variation. Direct comparison of the exact marker position was not possible due to the use of different marker systems in both cases; however, the positioning seems almost similar. Not many SR resistance QTLs were identified on 2DS until today. Bhavani et al. [45] identified an APR QTL on chromosome 2DS in a CIMMYT bi-parental population, PBW343/Kiritati, providing resistance to African *Pgt* pathotypes. Based on the closely linked marker *Xbarc095*, this QTL is positioned at around 14-15 Mb on the physical assembly [19]. *QSr.cim-2DS* mapped about 17–18 Mb based on DArTSeq markers on the physical assembly, which suggests that both of these QTLs could be similar, as both marker systems used were different for any direct comparison. Another 2DS QTL, *QSr.umn-2DS* [46], was mapped 14 cM away from the position of PBW343/Kiritati 2DS QTL, suggesting that this QTL is different to the QTL identified in the present study. The KASP marker *CIM0004* worked well and can be used for the selection of this QTL. 

### 3.4. QYrKen.cim-2DL

A QTL explaining 8–14% of the phenotypic variation in stripe rust response was detected on chromosome 2DL (*QYrKen.cim-2DL*) between the DArTSeq marker interval of 128–134 cM (~616 Mb) with a total length of 166 cM on the wheat consensus map of wheat v4. Until now, five genes have been located on chromosome 2D: *Yr8* [47], *Yr16* [48], *Yr37* [49], *Yr54* [27], and *Yr55* [50]. *Yr8* and *Yr37* are ASR genes and are different from *QYrKen.cim-2DL* because they confer different types of resistance. However, *Yr16* is an APR gene located around the centromeric region of the 2D chromosome [48,51], suggesting it was different from *QYrKen.cim-2DL*; however, it was difficult to obtain the relative physical position of *Yr55* for a precise comparison. *Yr54*, another APR gene on 2DL close to telomere, explained 49–54% of PVE [27]. The direct comparison was not possible, but the close positioning of *Yr54* to telomere and its strong phenotypic variation suggests it to be a different gene. 

Several other QTLs have also been reported on chromosome 2DL; *QPtst.jic.2D* was reported in a German cultivar, “Alcedo”, on 2DL [52], and, with overlapping markers and similar phenotypic variation with *Yr54*, it was speculated that they could be similar. *QYr.jki-2D* was mapped at the distal end of chromosome 2DL at around 636 Mb [53], whereas the QTL in our study was mapped at about 616 Mb of the physical assembly. Another minor QTL was reported on 2DL by Ren et al. [54] and was flanked by markers positioned between 513 and 608 Mb. This QTL was assumed to be linked to another minor QTL reported by Suenaga et al. [55] on chromosome 2DL. Therefore, these two minor QTLs might correspond to the QTL in the present study.

### 3.5. QYrKEN.cim-6BS

*QYrKEN.cim-6BS* was mapped between 21 and 24 Mb on chromosome 6BS. Two APR genes have been reported on chromosome 6BS: *Yr36* [56] and *Yr78* [57]. APR gene *Yr36* was mapped at 153 cM of the genetic map [58], whereas *Yr78* was mapped at 92.4 Mb on 6BS [57], suggesting that *QYrKEN.cim-6BS* can be different from *Yr36* and *Yr78*. Several other QTLs were reported on 6BS: *Qyr.sicau-6BS*, mapped between 80.4 Mb and 84.8 Mb on the physical assembly [59]; *Qyr.sun-6B*, mapped at the interval of about 150 Mb [60]; *QYrst.wgp-6BS.1*, mapped closer to *Yr36* on the basic SSR marker position [61]; *QYrsk.wgp*-*6B*, mapped to the centromere region of 6B [62]; and *QYr.cim-6BS*, considered to be *Yr78* based on the positioning of closely linked markers [30]. These results suggest that *QYrKEN.cim-6BS* could be different from previously reported genes/QTLs.

### 3.6. QSr.cim-3AL

A stem rust QTL, *QSr.cim-3AL*, was mapped between the DArTSeq marker interval of 65–68 cM of wheat consensus map v4. Not many QTLs have been reported on chromosome 3AL for stem rust. One QTL associated with stem rust was mapped on 3AL in the PBW343/Kenya Kudu bi-parental population around the marker interval of 73.4 cM [CIMMYT unpublished, [63]. The position suggests that this could be the same as *QSr.cim-3AL* in this study. Another 3AL QTL was also identified by association mapping of the durum panel [64] and was mapped around 85 cM, suggesting it to be a different QTL.

### 3.7. QSr.cim-6AS

The QTL *QSr.cim-6AS*, located on the short arm of chromosome 6A around 13 Mb, explained 8–15% of the phenotypic variation. Previously, *Sr8* locus alleles, *Sr8a* and *Sr8b*, have been reported on chromosome 6AS [65]. Several of the Ug99 pathotypes used on the phenotyping platform at KALRO, Njoro, are known to carry virulence to *Sr8a* [66,67], which rules out the possibility of this QTL being *Sr8a*. Another recent study mapped the *Sr8155b* gene on chromosome 6AS between 6.7 and 10.9 Mb [68], which is a possible allele of *Sr8*. A published KASP marker linked with *Sr8155b* was found to be monomorphic in the current study. Mokue#1 has been found to be resistant in field evaluations, even to the *Sr8155b* virulent pathotype, suggesting that the QTL could possibly be another allele of *Sr8*; this is under further characterization. Alternatively, it could be that the small effect seen is due to the low frequency of the *Sr8155b* virulent pathotype and higher frequency of the *Sr8155b* avirulent pathotype in the field.

## 4. Materials and Methods

### 4.1. Genetic Materials

A bi-parental F_5_-derived RIL population comprising 261 lines was developed from the cross between resistant parent Mokue#1 (Germplasm ID: 7396550; pedigree: Mucuy//Mutus*2/Tecue#1), a breeding line developed by CIMMYT in 2017, showing a high level of resistance against the three rusts and susceptible parent “Apav#1” (a RIL line selected from a cross of “Avocet-*YrA*” × “Pavon 76”, which is susceptible to YR, LR, and SR at all growth stages to predominant *Pst* and *Pt* pathotypes used in various trials conducted in Mexico and to *Pgt* in field trials in Kenya). 

### 4.2. Pathogen Materials

Predominant Mexican *Pst* and *Pt* pathotypes were used for testing the RIL population. For YR screening, *Pst* pathotype MEX14.191 having avirulence/virulence: *Yr1*, *4*, *5a*, *10*, *15*, (*17*), *24*, *26*, *5b*, *Poll*/*Yr2*, *3*, *6*, *7*, *8*, *9*, *27*, *31*, *A* [69] and MEX16.04 with additional virulence to *Yr10* and *Yr24* were used. The *Pt* pathotype MBJ/SP used in both seedling and field studies had the following avirulence/virulence: *Lr2a*, *2b*, *2c*, *3ka*, *9*, *16*, *19*, *21*, *24*, *25*, *28*, *29*, *30*, *32*, *33*, *36/1*, *3*, *3bg*, *10*, *11*, *12*, *13*, *14a*, *14b*, *15*, *17a*, *18*, *20*, *23*, (*26*), *27 + 31*, *37* [70]. For field phenotyping, pathotypes MEX14.191 and MBJ/SP were used to incite the disease epidemic. 

YR and SR field phenotyping was also carried out at Kenyan Agricultural Research and Livestock Organization (KALRO) research station in Njoro, Kenya, under natural conditions for YR, whereas for SR, a mixture of the prevalent Ug99 lineage pathotypes, TTKSK, TTKST, TTKTK, and TTKTT, was used [71].

### 4.3. Greenhouse Phenotyping for Stripe Rust and Leaf Rust

Seedling YR and LR phenotyping was carried out for the parents and the Mokue#1/Apav#1 RIL population in the greenhouse using *Pst* pathotype MEX 14.191 and MEX16.04, and *Pt* pathotype MBJ/SP, respectively. A YR differential set comprising 56 lines, mostly near-isogenic with known YR genes in the Avocet background, and another set of LR differentials (51 lines) with known LR genes in the Thatcher background, respectively, were also included in each phenotyping experiment. The parents, RILs, and differential sets were sown in plastic trays with 5-6 seeds/RIL and 30 RILs in a 29.5 × 23.5 cm tray with 3 cm distance between each RIL. Once the primary leaves were fully expanded, along with the emergence of the second leaf, the seedlings were inoculated in separate experiments with *Pst* pathotypes MEX14.191 and MEX16.04 and *Pt* pathotype MBJ/SP, by suspending urediniospores in a light mineral oil (Soltrol-170) and then atomizing them on seedlings. YR inoculated seedlings were incubated in a dew chamber for 24 h at 7–9 °C, whereas LR inoculated seedlings were incubated in a humidified chamber for 24 h to facilitate germination and infection processes. After 24 h, the YR and LR inoculated seedlings were transferred to microclimate growth rooms set at 17 ± 2 °C and 25 ± 2 °C, respectively. The assessment for YR and LR infection was carried out 12–14 days post inoculations using the 0–4 scale [12].

### 4.4. Field Phenotyping for Stripe Rust, Leaf Rust, and Stem Rust

#### 4.4.1. Stripe Rust and Leaf Rust Phenotyping in Mexico

Field phenotyping for YR resistance was carried out at CIMMYT research station in Toluca, Mexico (latitude 19.226581, longitude −99.551539, 2,640 masl) during the years 2020 and 2021 (YR20 and YR21). The LR experiments were conducted during 2018–2019 (LR19) at Norman E. Borlaug research station in Ciudad Obregon in the state of Sonora (latitude 27.33, longitude −109.93, 39 masl), and in 2020 and 2021 (LR20 and LR21) at CIMMYT research station in El Batan, Texcoco, Mexico (latitude 19.528192, longitude −98.84794, 2,249 masl). The RILs were sown on top of 80 cm wide raised beds as paired rows of 0.7 m length with a 0.3 m pathway. Infector rows, comprising a mixture of susceptible genotypes, were sown as hills on one side of the plot in the middle of the pathway, and around the field blocks to ensure uniform disease build-up on experimental plots. YR spreader mixture included “PBW343”, “Morocco”, “Murga”, “Nana”, six lines derived from the cross “Avocet/Atilla”, Avocet + *Yr24*, and Avocet + *Yr26*, whereas the spreader for LR included a mixture of “Morocco”, “Baart” “Sonalika”, “Nesser”, “Seri M 82”, “Sonora 64”, “Pitic S62”, “INIA F66”, and “Blanca Grande”. The *Pst* pathotype MEX14.191 and *Pt* pathotype MBJ/SP were used to inoculate the spreader rows after about 4–5 and 5–6 weeks of germination for YR and LR, respectively. The inoculation process was repeated three times. 

#### 4.4.2. Stripe Rust and Stem Rust Phenotyping in Njoro, Kenya

The parents and RIL population were phenotyped at the international rust screening platform, KALRO, Njoro, Kenya (latitude −0.341368, longitude 35.947650, 2,165 masl) for stem rust pathotypes (Ug99 variants) under conditions of artificial epidemics, and the phenotyping for stripe rust was performed under conditions of natural infection. The field plots comprised 0.7 m long paired rows sown 0.3 m apart in a flat field. A row was left empty between the plots and the alleyway was 0.3 m wide. Spreader rows comprised a mixture of susceptible cultivars and genotypes including “Cacuke”, “Eagle 10”, “Robin”, and six lines carrying the resistance gene *Sr24*. The spreader was planted as hill plots on one side of the paired rows in the middle of the alleyway and around the field block to facilitate uniform buildup and spread of the disease from booting to heading growth stages (Zadok’s growth stages 37–60). The spreader was artificially inoculated with a composite of East African *Pgt* pathotypes TTKSK, TTKST, TTKTK, and TTKTT having combined virulence to many resistance genes including *Sr24*, *Sr31*, *Sr38*, and *SrTmp*. YR pathotypes predominantly present at KALRO, Njoro, belonged to pathotype *Pst*S2 and *Pst*S11 [72] based on race analysis. However, the presence of other pathotypes in minor frequencies cannot be ruled out. 

#### 4.4.3. Rust Scoring

Rust responses were scored using a 0–100 disease rating scale according to the modified Cobb’s scale [20], whereas the infection type was assessed according to Roelfs et al. [73]. The first dataset was taken when the susceptible parental line reached 80% disease severity. The disease evaluation was repeated two to three times at 7–8 days interval from the first note taking. The repetition of data scoring is represented as “1” (first dataset), “2” (second data taken 7-8 days after first dataset), “3” (third data taken 14-16 days after first dataset), and “4” (fourth data taken about 21 days after first dataset), along with years 19 (2019), 20 (2020), and 21 (2021). Coefficient of infection (COI) values were generated by multiplying the stem rust severity value for each line by a constant value for each infection response: R = 0.2, RMR = 0.3, MR = 0.4, M = 0.6, MS = 0.8, S = 1.0 [74]. Both mean disease severity and average coefficient of infection of the datasets were used for QTL analyses.

### 4.5. Statistical Analysis

The strength of linear correlation of mean disease severity (MDS) across various years and locations was calculated using Pearson’s correlation coefficient. Genotypic and phenotypic segregation data from the RIL population were subjected to chi-squared (χ^2^) analysis to test goodness-of-fit to expected genetic ratios. The number of loci segregating for reaction to rust in the RIL population grown under field conditions was estimated according to the following [75]:*n =* (*GR*)^2^/4.57(*σ*^2^*g*)
where *n* is the minimum number of segregating loci, *GR* is the genotypic range = phenotypic range × h (narrow-sense heritability, h = *σ*^2^*g*/*σ*^2^*g* + *σ*^2^*e*), *σ*^2^*g* is the genetic variance of the RILs, and 4.57 is a correction factor for inbreeding at F_5_.

### 4.6. Genotyping

#### 4.6.1. DNA Extraction, Quantification, and DArTSeq Genotyping

Genomic DNA was extracted from 10–12-day-old seedlings of the RIL population and parental lines, Mokue#1 and Apav#1, according to the CIMMYT-optimized CTAB protocol [76]. After extraction, DNA quality and quantity were checked using a Nanodrop 8000 spectrophotometer (Thermo Scientific, Waltham, Massachusetts, USA). The DNA samples of 261 RILs and parental lines were subjected to Diversity Arrays Technology Sequencing (DArTSeq) platform through SAGA in the CIMMYT Biotech Laboratory. 

#### 4.6.2. Marker Validation

A previously reported KASP marker for genes, *Lr46*/*Yr29*, was validated on the RIL population. KASP genotyping was performed using the protocol described in Qureshi et al. [77]. Additionally, for the 2NS/2AS region, two STS markers, *WGGB156* [40] and *cslVrgal3*, which were developed from a follow-up study of Seah et al. [78] and He et al. [41], and one KASP marker, *Yr17*/*Lr37*/*Sr58*, were used [42]. 

#### 4.6.3. Linkage Map Construction and QTL Analysis

The linkage map of the Mokue#1/Apav#1 RIL population was constructed using the ASMap package in R software according to the protocol mentioned in Hussain et al. and Taylor et al. [79,80]. To maximize the accuracy of mapping data, both QTL Cartographer v2.5 [81] and QTL IciMapping 4.1 [82] based on the inclusive composite interval mapping (ICIM) model were performed using 500–1000 permutations. A QTL was only considered significant when the LOD score was ≥3. Percentage phenotypic variance explained (PVE) was estimated using the stepwise regression at the LOD peaks. Genetic map/figures for the QTLs were prepared using MapChart software [83].

## 5. Conclusions

Frequent detection of new pathotypes of wheat rust fungi demands continuous efforts for the discovery and characterization of new sources of resistance, along with development of the tools for their incorporation into adapted wheat germplasm. CIMMYT’s advanced breeding line “Mokue#1” showed high levels of resistance to YR and LR in Mexico, and to YR and SR in Kenya. Therefore, this study was conducted to characterize these resistance sources, which led to the identification of three consistent pleiotropic QTLs on chromosomes 1BL, 2AS, and 2DL, along with two minor QTLs on chromosomes 2DL and 6BS for YR and two minor QTLs on 3AL and 6AS for SR. These QTLs showed a significant reduction in disease severity when present in combination. To attain durable resistance, gene pyramiding has been the way forward and the combination of genes/QTLs present in “Mokue#1” can be used in breeding programs for their transfer to enhance durability.

## Figures and Tables

**Figure 1 ijms-24-12160-f001:**
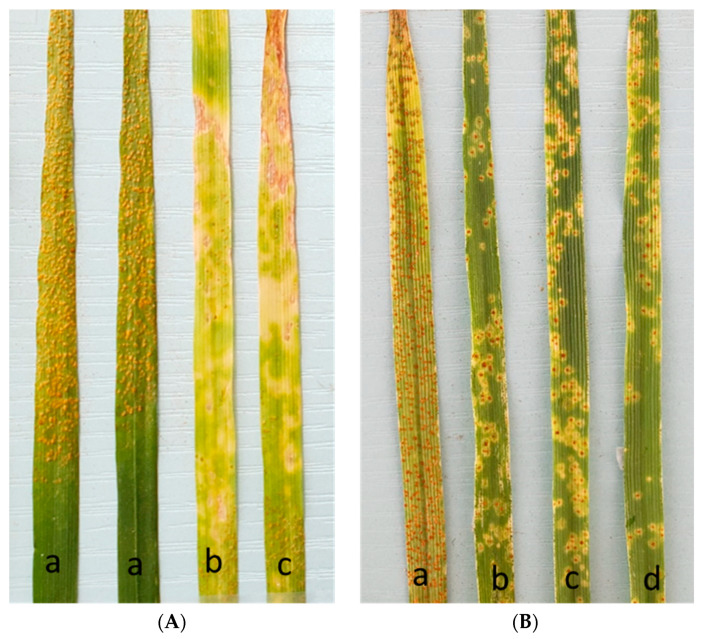
(**A**) Seedling infection type (IT) responses to *Puccinia striiformis* race MEX14.191: (a) susceptible parent showing IT 4, (b) resistant parent showing IT 2C, (c) homozygous resistant RIL (IT 23CN). (**B**). Seedling responses to *Puccinia triticina* race MBJ/SP: (a) susceptible parent showing IT 4, (b) resistant parent showing IT of 12C, (c, d) RILs showing IT 1C.

**Figure 2 ijms-24-12160-f002:**
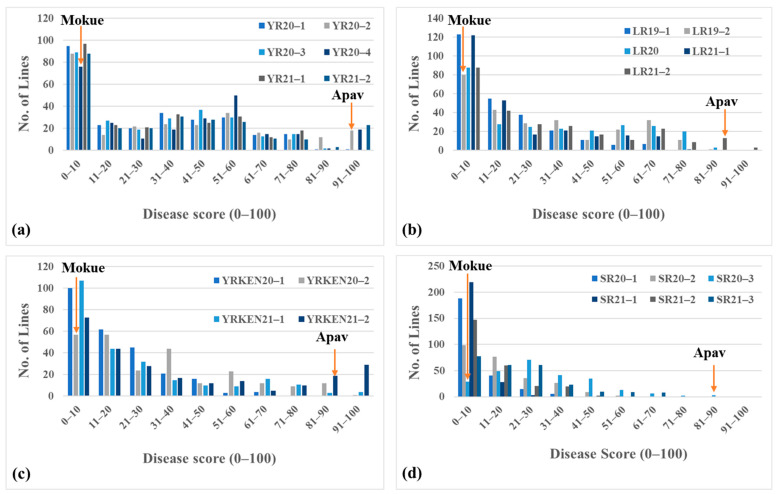
Frequency distribution of Mokue/Apav#1 RIL population against (**a**) stripe rust under field conditions in Toluca, Mexico; (**b**) leaf rust under field conditions in El Batan and Obregon, Mexico; (**c**) stripe rust under field conditions in Njoro, Kenya; and (**d**) stem rust under field conditions in Njoro, Kenya.

**Figure 3 ijms-24-12160-f003:**
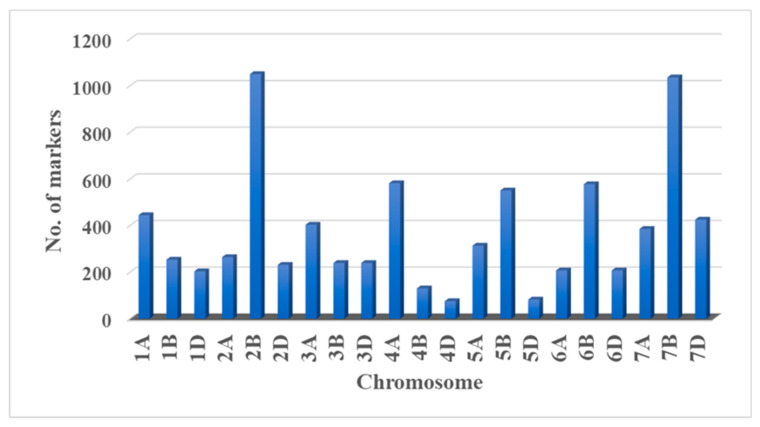
Distribution of markers across A, B, and D genomes.

**Figure 4 ijms-24-12160-f004:**
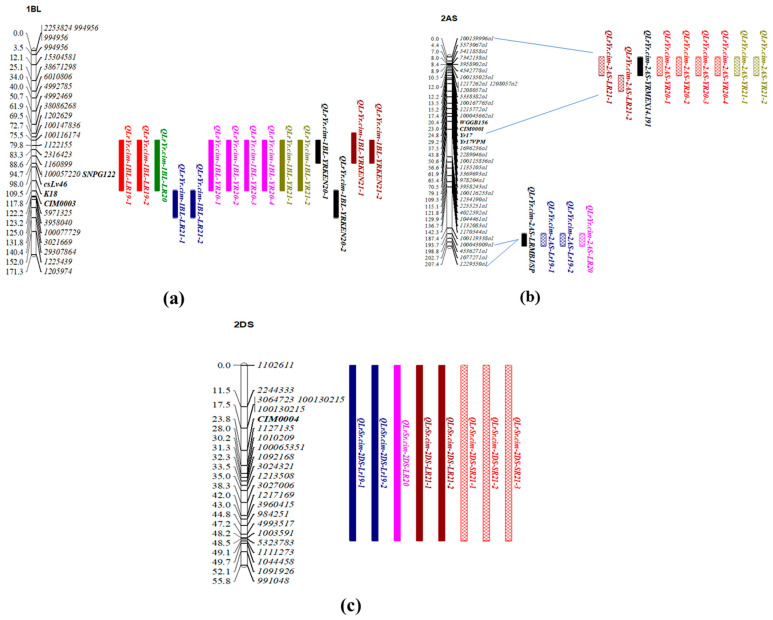
Genetic maps of chromosomes associated with pleiotropic loci identified in the Mokue/Apav#1 RIL population: (**a**) *QLrYr.cim-1BL* on chromosome 1BL, (**b**) *QLrYr.cim-2AS* on chromosome 2AS, and (**c**) *QLrSr.cim-2DS* on chromosome 2DS.

**Figure 5 ijms-24-12160-f005:**
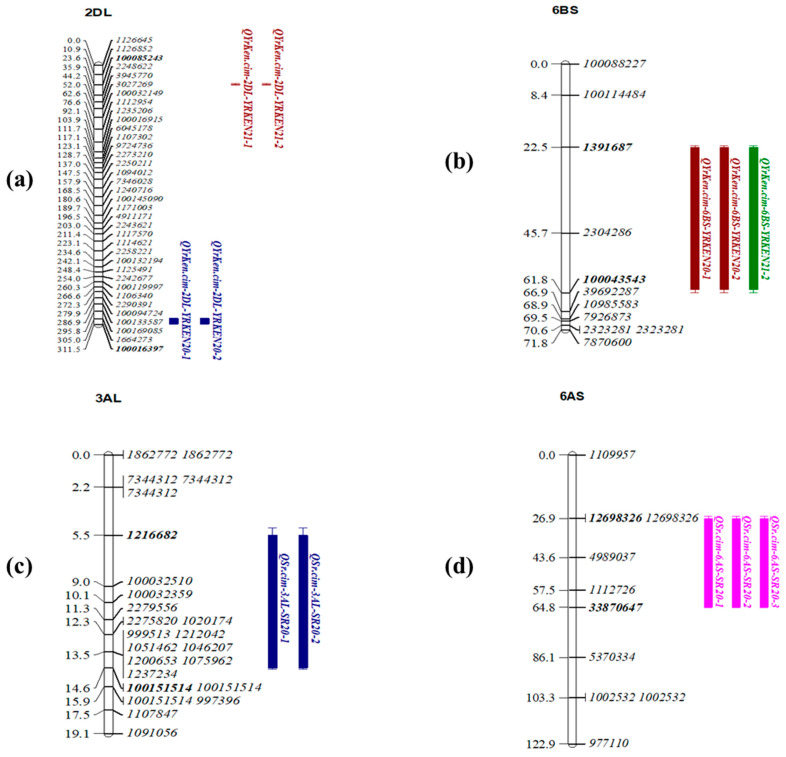
Genetic maps of chromosomes associated with minor effect QTLs in the Mokue/Apav#1 RIL population: (**a**) *QYrKen.cim-2DL*, (**b**) *QYrKen.cim-6BS*, (**c**) *QSr.cim-3AL*, and (**d**) *QSr.cim-6AS*.

**Figure 6 ijms-24-12160-f006:**
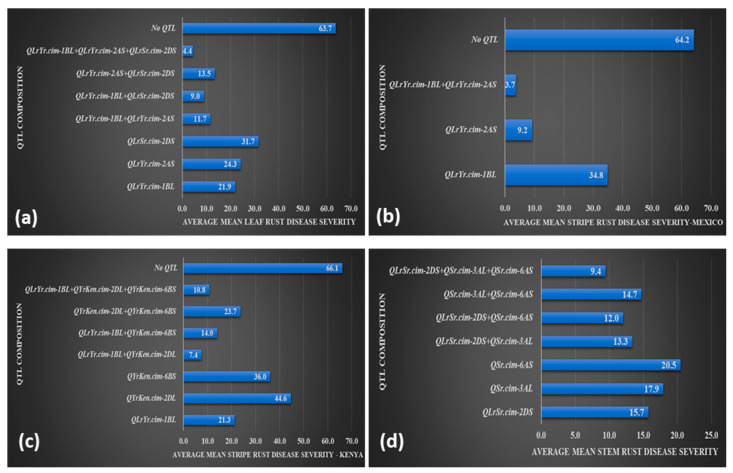
Mean disease severity (MDS) of lines carrying different combinations of QTLs. Lines containing the different QTL combinations were grouped together and the corresponding rust severities were averaged over environments: (**a**) leaf rust combinations, (**b**) stripe rust combinations from Mexico data, (**c**) stripe rust combinations based on Kenya data, and (**d**) stem rust combinations based on Kenya data.

**Table 1 ijms-24-12160-t001:** Pearson correlation coefficients (r) for two-way comparisons of leaf rust and stripe rust severity data from different environments in Mexico.

	LR19-1	LR19-2	LR20	LR21-1	LR21-2	YR20-1	YR20-2	YR20-3	YR20-4	YR21-1
LR19-2	0.82									
LR20	0.69	0.80								
LR21-1	0.78	0.84	0.81							
LR21-2	0.79	0.85	0.82	0.94						
YR20-1	0.57	0.65	0.61	0.65	0.67					
YR20-2	0.59	0.69	0.64	0.69	0.71	0.97				
YR20-3	0.55	0.65	0.63	0.65	0.67	0.92	0.93			
YR20-4	0.56	0.67	0.64	0.67	0.69	0.92	0.94	0.97		
YR21-1	0.53	0.60	0.62	0.62	0.66	0.85	0.86	0.87	0.88	
YR21-2	0.57	0.62	0.62	0.66	0.68	0.85	0.87	0.87	0.88	0.97

All correlations are significant at *p* < 0.01. LR indicates leaf rust; YR indicates stripe rust. The numbers 19, 20, and 21 indicate the years 2019, 2020, and 2021, respectively. After the year, 1, 2, 3, and 4 represent first, second, third, and fourth dataset, respectively, taken within a week interval.

**Table 2 ijms-24-12160-t002:** Pearson correlation coefficients (r) for two-way comparisons of stripe rust and stem rust severity data from Njoro, Kenya.

	YRKEN20-1	YRKEN20-2	YRKEN21-1	YRKEN21-2	SR20-1	SR20-2	SR20-3	SR21-1	SR21-2
YRKEN20-2	0.93								
YRKEN21-1	0.86	0.88							
YRKEN21-2	0.86	0.88	0.96						
SR20-1	0.30	0.31	0.24	0.18					
SR20-2	0.22	0.22	0.14	0.10	0.91				
SR20-3	0.16	0.17	0.08	0.06	0.80	0.91			
SR21-1	−0.06	−0.07	−0.10	−0.13	0.51	0.53	0.55		
SR21-2	−0.14	−0.14	−0.18	−0.22	0.51	0.55	0.59	0.88	
SR21-3	−0.16	−0.16	−0.20	−0.24	0.48	0.53	0.59	0.79	0.93

All correlations are significant at *p* < 0.01. LR indicates leaf rust; YR indicates stripe rust. The numbers 19, 20, and 21 indicate the years 2019, 2020, and 2021, respectively. After the year, 1, 2, 3, and 4 represent first, second, third, and fourth dataset, respectively, taken within a week interval.

**Table 3 ijms-24-12160-t003:** Summary of stripe rust, leaf rust, and stem rust QTLs identified in Mokue/Apav RIL population.

QTL	Year *	Left Peak Marker	Right Peak Marker	Wheat Consensus Map v4	LOD	PVE ** (%)	Additive Effect
*QLrYr.cim-1BL*	LR19-1	*100116174a1*	*CIM0003*	257–258 cM	18.5	18	7.1
	LR19-2	*100116174a1*	*CIM0003*	257–258 cM	18.86	17	9.5
	LR20	*100116174a1*	*CIM0003*	257–258 cM	11.02	15	10.4
	LR21-1	*CIM0003*	*2930786a1*	257–258 cM	5.19	5	6.84
	LR21-2	*CIM0003*	*2930786a1*	257–258 cM	5.74	5	8.24
	YR20-1	*100116174a1*	*CIM0003*	257–258 cM	17.42	10	8.46
	YR20-2	*100116174a1*	*CIM0003*	257–258 cM	18.14	11	10.2
	YR20-3	*100116174a1*	*CIM0003*	257–258 cM	15	9	7.6
	YR20-4	*100116174a1*	*CIM0003*	257–258 cM	17.9	10	9.7
	YR21-1	*100116174a1*	*CIM0003*	257–258 cM	10.2	5	6
	YR21-2	*100116174a1*	*CIM0003*	257–258 cM	14.3	8	9.15
	YRKEN20-1	*100116174a1*	*SNPG122*	257–258 cM	21.8	22	7.6
	YRKEN20-2	*CIM0003*	*2930786a1*	257–258 cM	20.64	19	10.9
	YRKEN21-1	*1202629a1*	*SNPG122*	257–258 cM	14.97	17	10.2
	YRKEN21-2	*1202629a1*	*SNPG122*	257–258 cM	13	13	11.9
*QLrYr.cim-2AS*	LRMBJ/SP	*100043009*	*1229550*	8–15 cM	5.44	6	0.12
	Lr19-1	*100043009*	*1229550*	8–15 cM	15.5	15	6.39
	Lr19-2	*100043009*	*1229550*	8–15 cM	21.96	19	10.92
	LR20	*100043009*	*1229550*	8–15 cM	19.3	19	11.65
	LR21-1	*100159996*	*WGGB156*	3–8 cM	16.58	13	7.51
	LR21-2	*WGGB156*	*1696236*	3–8 cM	28.6	24	13.6
	YRMEX14.191	*100159996*	*WGGB156*	3–8 cM	14.72	21	0.23
	YR20-1	*100159996*	*WGGB156*	3–8 cM	59	46	18.7
	YR20-2	*100159996*	*WGGB156*	3–8 cM	55.8	45	22.1
	YR20-3	*100159996*	*WGGB156*	3–8 cM	12.6	7	11.2
	YR20-4	*100159996*	*WGGB156*	3–8 cM	13.6	7	13.6
	YR21-1	*100159996*	*WGGB156*	3–8 cM	14.2	7	11.4
	YR21-2	*100159996*	*WGGB156*	3–8 cM	4.7	3	9.25
*QLrSr.cim-2DS*	Lr19-1	*1102611*	*1111273*	14–16 cM	14	14	6.3
	Lr19-2	*1102611*	*1111273*	14–16 cM	18.8	15	9.4
	LR20	*1102611*	*1111273*	14–16 cM	15.2	14	9.9
	LR21-1	*1102611*	*1111273*	14–16 cM	19.9	19	8.6
	LR21-2	*1102611*	*1111273*	14–16 cM	24.7	20	11.5
	SR21-1	*1102611*	*1111273*	14–16 cM	6.43	9	2
	SR21-2	*1102611*	*1111273*	14–16 cM	8.7	12	4
	SR21-3	*1102611*	*1111273*	14–16 cM	7	8	4.3
*QYrKen.cim-2DL*	YRKEN20-1	*100016397*	*NA* ***	134 cM	8.65	8	4.6
	YRKEN20-2	*100016397*	*NA* ***	134 cM	11.6	11	8.1
	YRKEN21-1	*100085243*	*NA* ***	128 cM	12.11	13	9.2
	YRKEN21-2	*100085243*	*NA* ***	128 cM	14.4	15	13.2
*QYrKen.cim-6BS*	YRKEN20-1	*1391687*	*100043543*	21–24 Mb ****	6.97	8	4.3
	YRKEN20-2	*1391687*	*100043543*	21–24 Mb ****	4	4	4.6
	YRKEN21-2	*1391687*	*100043543*	21–24 Mb ****	4	5	7.18
*QSr.cim-3AL*	SR20-1	*1216682*	*100151514*	65–68 cM	5.6	8	2.64
	SR20-2	*1216682*	*100151514*	65–68 cM	9.07	13	6.05
*QSr.cim-6AS*	SR20-1	*12698326*	*33870647*	13 Mb ****	4	8	3
	SR20-2	*12698326*	*33870647*	13 Mb ****	7.4	14	5.04
	SR20-3	*12698326*	*33870647*	13 Mb ****	7.3	15	6.48

* Year = The number in this column represents years, i.e., 19 = 2019, 20 = 2020, 21 = 2021. After the year, 1, 2, 3, and 4 represent first, second, third, and fourth dataset taken within a week interval; ** PVE = phenotypic variation; *** NA = not applicable; **** The marker position was not available in cM on wheat consensus map v4 so Chinese Spring IWGSC v2.0 reference positions were used.

**Table 4 ijms-24-12160-t004:** List of KASP markers used in this study.

S/No.	DArTSeq Marker	KASP Marker	Allele A1 ^a^ Primer	Allele A2 ^b^ Primer	Common Primer	SNP
1	*2259918*	*CIM0003*	tcaaacactcgtcacagtacc	tcaaacactcgtcacagtacg	ctcgaacatcacgtcctccc	[G/C]
2	*100072906*	*CIM0001*	tgggcgtgaagatggagaa	tgggcgtgaagatggagag	ctccaggcaggggagctc	[T/C]
3	*991084*	*CIM0004*	atgtccggacagactgcagg	atgtccggacagactgcaga	ccctctgagcaagcatacga	[G/A]

^a^ A1 primer labeled with FAM: GAAGGTGACCAAGTTCATGCT; ^b^ A2 primer labeled with HEX: GAAGGTCGGAGTCAACGGAT.

## Data Availability

Not applicable.

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
