# Peer review of "Genomic Regions Associated with Resistance to Three Rusts in CIMMYT Wheat Line “Mokue#1”"

_ijms, 2023, doi:10.3390/ijms241512160_

Round 1
Reviewer 1 Report
Review of the manuscript titled: Genomic Regions Associated with Resistance to Three Rusts in CIMMYT Wheat Line “Mokue#1”
The manuscript is devoted to an important subject of wheat resistance to three rusts and uses unique genotype to find the genetic basis of resistance. The methodology and the tools used for the study are relevant. The phenotyping was conducted in the greenhouse and under field conditions and provided sufficient data for GWAS. The manuscript is well written, provides important contribution to the knowledge of wheat disease resistance and deserves publishing. There are a couple of comments which may be considered.
Broader information on the Mokue#1 in terms of adaptation, yield, grain quality, etc would help in potential use of this line by cooperators and researchers.
The designations of LR19-1, LR19-2, YR20-1, YR20-2 etc are not well explained in the methods and perhaps refer to the date of evaluation. Also LR19 may be confused with respective gene especially in the tables.
Reviewer 2 Report
Manuscript "Genomic Regions Associated with Resistance to Three Rusts in CIMMYT Wheat Line "Mokue#1" by Qureshi and co-authors is dedicated to the serious and up-to-date problem of searching for new genes or QTLs conferring resistance to yellow, black, and brown rusts. The authors designed a study involving both green house and multi-year and multi-locus phytopathological evaluation of the F5 RILs derived from CIMMYT Wheat Line "Mokue#1". The conclusions of the research are based on the clearly demonstrated results that have high significance for marker-assisted breeding of resistant cultivars of wheat.
However, there are some inconsistencies that should be corrected or explained. The major issue is the scheme of the field experiments. In Table 1, you show LR19-1 LR19-2 LR20 LR21-1 LR21-2 YR20-1 YR20-2 YR20-3 YR20-4 YR21-1 YR21-2, in Table 2, you show YRKEN20-1 YRKEN20-2 YRKEN21-1 YRKEN21-2 SR20-1 SR20-2 SR20-3 SR21-1 SR21-2 SR21-3 with no deciphering of the abbreviations. It is clear that LR, YR and SR stand for rust, and 19, 20, and 21 stand for years, but "-1", "-2", "-3", and "-4" remain unclear. You should explain it and include their meaning in Methods if they mean different sites. The geographical coordinates of field experimental stations should also be shown.
The second principle question is the level of significance that was used in the Chi-squared (χ2) analysis; it should be indicated in Section 2.5.
The third unclarity is the sentence in lines 202 and 212: "The segregating lines were averaged out and added into the respective categories." What did you mean? Please decipher this and describe it in more detail in Methods.
Warm regards,
Reviewer.
English should be improved, as too many typos and mistakes are found at a brief glance. The following typos and improper text formatting have been found:
Different types of quotes are used for Mokue#1: "Mokue#1" in the title and abstract (line 13), ‘Mokue#1’ in the overall text (line 75), or no quotes at all (line 84). Please make it uniform.
lines 61–64: the gene symbols should be in italics.
line 77: "This study was conducted to; " colon should be used instead of a semicolon.
line 83: "261lines", line 147 "TTKTTwith",line 279 "markers1202629a1", line 433 "Ptpathotype" - spaces missed
line 205: "Puccinia striiformis", line 207 "Puccinia triticina" - should be in italics.
Figures 2, 4, and 5's resolution should be increased. In addition, please use another font color than yellow; it is invisible for reading.
Tabe 2: The line between YR20-3 100159996 and YR20-4 100159996 in the QLrYr.cim-2AS box should be removed.
line 322: "minor effect LR gene" - probably, you mean "Lr gene"?
line 444: "However, two STS markers." -a comma should be used instead of a full stop.
line 451: "Pt pathotype" - Pt should be in italics
You use "in-house" in lines 287 and 450 and "inhouse" in lines 316 and 446. Please make it uniform.
